# Role of High-Mobility Group Box-1 in Liver Pathogenesis

**DOI:** 10.3390/ijms20215314

**Published:** 2019-10-25

**Authors:** Bilon Khambu, Shengmin Yan, Nazmul Huda, Xiao-Ming Yin

**Affiliations:** Department of Pathology and Laboratory Medicine, Indiana University School of Medicine, Indianapolis, IN 46202, USA; yan29@iupui.edu (S.Y.); nhuda@iu.edu (N.H.); xmyin@iupui.edu (X.-M.Y.)

**Keywords:** HMGB1, autophagy, inflammation, fibrosis, ductular reaction, tumor, NAFLD, ALD, DILI

## Abstract

High-mobility group box 1 (HMGB1) is a highly abundant DNA-binding protein that can relocate to the cytosol or undergo extracellular release during cellular stress or death. HMGB1 has a functional versatility depending on its cellular location. While intracellular HMGB1 is important for DNA structure maintenance, gene expression, and autophagy induction, extracellular HMGB1 acts as a damage-associated molecular pattern (DAMP) molecule to alert the host of damage by triggering immune responses. The biological function of HMGB1 is mediated by multiple receptors, including the receptor for advanced glycation end products (RAGE) and Toll-like receptors (TLRs), which are expressed in different hepatic cells. Activation of HMGB1 and downstream signaling pathways are contributing factors in the pathogenesis of non-alcoholic fatty liver disease (NAFLD), alcoholic liver disease (ALD), and drug-induced liver injury (DILI), each of which involves sterile inflammation, liver fibrosis, ductular reaction, and hepatic tumorigenesis. In this review, we will discuss the critical role of HMGB1 in these pathogenic contexts and propose HMGB1 as a bona fide and targetable DAMP in the setting of common liver diseases.

## 1. HMGB1 is a Prototypical Damage-Associated Molecular Pattern Molecule

Damage-associated molecular pattern (DAMPs), also known as amphoterin, danger signals, or alarmins, are host intracellular (cytosolic or nuclear) biomolecules that are secreted, released, or exposed on the cellular surface by stressed or dying cells. DAMPs generally communicate an endogenous danger signals to the neighboring cells. DAMP could be non-protein such as ATP, Uric acid, heparin sulfate, or protein DAMP such as HMGB1, S100A8, S100A9, Histones, and Serum Amyloid A (SAA). Among the protein DAMPs, HMGB1 is the best-studied DAMP.

High-mobility group box1 (HMGB1) is a key chromosomal non-histone protein regulating DNA structure by binding to and bending through the minor groove. Human HMGB1 consists of 215 amino acid residues and has two L-shaped DNA-binding domains (HMG A box (9–79aa), HMG B box (95–163aa)) and a shorter C-terminal tail (186–215aa) [1]. Structurally, HMGB1 shares 99% amino acid sequence identity between rodents and humans [2,3]. HMGB1 is ubiquitously expressed at a very high level in both parenchymal and non-parenchymal hepatic cells (Figure 1), which is believed to be only 10 times less abundant than the core histones. HGMB1 is considered as a prototypical DAMP because it has well-defined intracellular roles in the absence of cellular stress and can also initiate and perpetuate a non-infectious (i.e., sterile) inflammatory responses, alone or in combination with other molecules, such as DNA, when released by stress or dying cells. The first evidence of HMGB1 as the prototypical DAMP was shown by Tracey and collaborators [4]. HMGB1 was reported to be actively secreted by mouse macrophages as a late mediator of endotoxin lethality [4]. HMGB1 was subsequently identified to be released by necrotic and apoptotic cells [5,6].

## 2. HMGB1 Has Multiple Functions Depending on Its Cellular Location

In normal conditions, hepatic cells express HMGB1 abundantly in the nucleus, whilst low in the cytoplasm. Interestingly, HMGB1 is not present or is significantly reduced in the nucleus of some hepatic cells

(Figure 1) [7]; the reason is unknown to date. Due to the presence of two nuclear localization sequences, newly synthesized HMGB1 is mainly translocated into the nucleus for its nuclear architectural function. Further, post-translational modification through the redox regulation of critical cysteine residues plays a key role in the inter- and intra-cellular translocation of HMGB1 (see recent reviews [8,9] for detail). Depending on the location of HMGB1, inside and outside the cells, its function varies. HMGB1 is also expressed in the cell surface in mononuclear immune cells [10], neuronal cells [11], and tumor cells [12]. Cell surface-specific expression of HMGB1 could be related to the active secretion process.

### 2.1. Nuclear HMGB1 Function as DNA Chaperone

In the nucleus, HMGB1 acts as a DNA chaperone, by binding to DNA structure without sequence specificity (through the minor groove) in a transient manner (not stably) and bending the DNA reversibly [13]. Nuclear HMGB1 is chemically non-acetylated and in a reduced (thiol) form. Generally, nuclear HMGB1 acts as an “architectural” factor to facilitate the assembly of certain nucleoprotein complexes and participates in fundamental events such as DNA replication, remodeling, and DNA repair. The B-box structural motif of HMGB1 mainly contributes to its DNA bending function of HMGB1 [14].

HMGB1 is also well known to play a role in gene transcription. The well-known transcription role of HMGB1 is in the regulation of glucocorticoid receptor expression. HMGB1-ablated newborn mice die from hypoglycemia due to the impairment of the activation of gene expression by the glucocorticoid receptor [15]. Interestingly, conditional ablation of *Hmgb1* in specific cell subsets in the liver, such as in hepatocytes, does not cause a deleterious phenotype suggesting that the HMGB1 role is more important during the embryonic development and its function could be substituted by some other protein factors possibly other HMGB proteins [16].

### 2.2. Cytosolic HMGB1 Regulates Autophagy

Under a physiological stress condition such as starvation, HMGB1 could translocate into the cytosol due to elevated reactive oxygen species (ROS). Cytosolic HMGB1 can then interact with the BECLIN1–BCL2 protein complex to release BECLIN1, which can then induce autophagy [17]. Under pathological conditions such as during tumorigenesis, HMGB1 has been reported to interact with the p53 inside the nucleus. Genetic deletion of p53 can then cause HMGB1 translocation into the cytosol to regulate autophagy and apoptosis [18].

Cytosolic HMGB1 has also been shown to prevent the protective autophagy proteins BECLIN1 and ATG5 from calpain-mediated cleavage during inflammation and hence prevents apoptotic injury by inducing pro-survival autophagy pathway [19]. On the other hand, cytosolic HMGB1 has also been demonstrated to sustain cellular bioenergetics and mitochondrial morphology in non-hepatic cells in vitro [20]. Intracellular HMGB1 functions as a transcriptional regulator of heat shock protein beta-1 (HSPB1) gene expression. HSPB1 could, in turn, regulate a selective form of autophagy called mitophagy by regulating the cellular actin cytoskeleton [20]. Loss of HMGB1 results in a mitophagy defect characterized by mitochondrial fragmentation, decreased aerobic respiration, and subsequent ATP production (defective oxidative phosphorylation). In contrast, liver-specific genetic loss of *Hmgb1* does not alter the mitochondrial structure and function in vivo [16]. Moreover, both general autophagy and mitophagy occurred normally in the absence of HMGB1. This suggests that hepatic HMGB1 is dispensable for autophagy and mitochondrial quality control, at least in the in vivo situation [16]. Interestingly, hepatic autophagy is also required for the HMGB1 release (see below).

### 2.3. Extracellular HMGB1 Functions as an Alarmin

Extracellular HMGB1 (either passively released or actively secreted) triggers inflammation and adaptive immunological responses by switching among multiple oxidative states. Extracellular HMGB1 interacts with binding partners such as receptor for the advanced glycation end product (RAGE), Toll-like receptors (TLRs), or other receptors such as CD24/Siglec, Syndecans, and Mac-1 [21]. RAGE and TLR4 are two of the most prevalent and well-studied HMGB1 extracellular receptors. HMGB1 signals through RAGE and TLR4 in numerous cell types to activate various signaling pathways such as p38/p42/44 MAPK, JNK, MEK1/2, ERK1/2, MyD88, NF-kB, and PI3K-AKT [22,23,24,25]. How HMGB1 could activate such a diverse signaling pathway is less clear. It is possible that the various forms of post-translational modification including acetylation, methylation, phosphorylation, and redox modify the HMGB1 capacity to modulate these signaling pathways. The downstream pathophysiological impacts of HMGB1 mediated activation of these signaling pathways are discussed below in detail. Extracellularly, HMGB1could also bind to non-receptor proteins such as thrombomodulin (a regulator of coagulation) [26] and haptoglobin (an acute-phase protein) [27]. These non-receptor proteins scavenge extracellular HMGB1, thereby reducing its inflammatory signals.

## 3. HMGB1 Release Depends on the Nature of Cellular Stress

HMGB1 is a non-histone nuclear protein normally residing in the nucleus. Upon cellular stress or tissue injury, HMGB1 could be post-translationally modified via acetylation, phosphorylation, methylation, or oxidation [8]. These modifications not only modulate HMGB1 structure, localization, and biological functions, but also its release from the cells. Notably, HMGB1 could also be structurally modified during its release process. For example, HMGB1 can be converted into an oxidized form (disulfide-HMGB1) by the oxidative extracellular environment [28].

HMGB1 is classically known to be passively released from the dead cells and actively secreted as a soluble protein by severely stressed cells. HMGB1 is considered as the “leaderless protein” as it lacks the conventional N-terminal hydrophobic secretion signal peptide [29]. Hence, HMGB1 is not secreted by a conventional secretory pathway, which means it is not directly translocated from the Golgi apparatus to the extracellular space. How exactly HMGB1 is secreted is unclear. The current proposed model suggests that the HMGB1 release is initiated by the nucleo-cytoplasmic shuttling of HMGB1 following a post-translational modification, redirecting the molecule moving towards cytosol for extracellular release.

HMGB1 release can occur rapidly and passively during the course of necrosis and apoptosis. During apoptotic cell death, HMGB1 is confined within the cleaved DNA fragment and enclosed within the apoptotic bodies whereas, in the necrotic form of cell death, HMGB1 is only weakly adherent to the chromatin and, thus, can readily exist from cells with the disruption of membrane permeability. Thus, the extracellular release of HMGB1 through necrotic or apoptotic mode could have a different downstream effect. For example, HMGB1 release during apoptosis is non-inflammatory (tolerogenic), in contrast to its release during necrosis process, which could trigger inflammation [5,30].

HMGB1 can also be actively secreted by different types of cell such as monocytes, macrophages, dendritic cells, natural killer cells, endothelial cells, and tumor cells [4]. The earliest event of the active cellular secretion appears to be a posttranslational modification (acetylation) of HMGB1 in the cytosol, which prevents it from getting into the nuclear compartment, hence increasing its cytosolic retention. In contrast, the deacetylation of HMGB1 increases its nuclear retention. Recently, the Sirtuin 1, silent information regulator 2-related enzyme 1-mediated deacetylation of HMGB1 has been demonstrated in lipopolysaccharide (LPS)-induced murine macrophages [31]. Sirtuin 1 directly interacts with HMGB1 via its *N*-terminal lysine residues [28,29,30] to form a stable complex with HMGB1 and thereby inhibited HMGB1 release via deacetylation. Deacetylation of HMGB1 can also be mediated by histone deacetylase (HDAC) 1/4 to prevent the HMGB1 active release in ischemic-reperfused (IR) liver [32]. Hence, reduction in nuclear HDAC1/4 activity causes elevated acetylated-HMGB1 in the serum during IR injury [32]. Thus, hyperacetylation shifts its equilibrium from a predominantly nuclear location towards cytosolic accumulation. The cytosolic HMGB1 could then be loaded into the secretory lysosomes or another vesicular compartment such as the exosome or the microvesicle, from which it can be secreted. However, the precise understanding of how cytosolic HMGB1 is loaded into these secretory vesicles and what molecular factors are involved in the HMGB1 loading process remain to be described.

Active release of HMGB1 has also been recently reported for the autophagy-deficient hepatocytes [7]. We will next discuss below the active release process in the context of autophagy-deficient hepatocyte, which could reflect the scenario of HMGB1 release by the stressed hepatocyte.

## 4. Hepatic Autophagy Inhibition Increases the Active Release of HMGB1

Autophagy is an intracellular lysosomal degradative pathway required for liver homeostasis. Classically, autophagy is well known for its role in metabolic functions where it provides nutrients for survival in response to stress condition. Recently, the autophagy process and the autophagy-related proteins have been shown to be important for the secretion of diverse intracellular proteins involved in intercellular communication [33]. Hence, the term “secretory autophagy” has been used to describe the process in which the canonical autophagy pathway takes part in the secretion of proteins by transporting them in the autophagosomes directly to the plasma membrane, to multivesicular bodies, or to secretory lysosomes for their extracellular release [33].

While activation of autophagy has been found to be involved in increased protein secretion, such as IL-1β in lipopolysaccharide (LPS)-stimulated macrophage cells [34], enhanced protein secretion has also been reported due to inhibition of autophagy [35]. For example, inhibition of autophagy increased the secretion of proinflammatory cytokines such as macrophage migration inhibitory factor (MIF) from LPS-activated macrophages [35]. Whether autophagy machinery is used for secretory function or autophagy indirectly modulates protein secretion is unknown.

Inhibition of hepatic autophagy also increases secretion of hepatic protein factors such as HMGB1. In a genetic model of hepatic autophagy deficiency targeting the parenchymal cell, we have found that HMGB1 is actively released independently of cellular injury [7]. In an inducible *Atg7* knocked-out mouse strain, HMGB1 is found to be released almost immediately after *Atg7* deletion, before the development of liver injury. Moreover, HMGB1 release occurs even in the proliferating but autophagy-incompetent hepatocytes. Mechanistically, autophagy-deficient hepatocytes are found to activate the anti-oxidative stress response-related transcription factor Nuclear factor erythroid 2-related factor 2 (NRF2), which causes the release of HMGB1. Genetic ablation of NRF2 in the autophagy-deficient liver suppresses HMGB1 release. Consistent with this observation, pharmacological or genetic activation of NRF2 alone, without disabling autophagy or causing injury, could cause HMGB1 release. Furthermore, activation of NRF2 was associated with activation of Caspase 1/11 inflammasome for the HMGB1 release, as genetic disruption of Caspase-1/11 blocked the HMGB1 release (Figure 2) [7]. Finally, the NRF2-mediated Caspase1/11 activation is correlated with the cleavage of gasdermin D (GSDMD), which could oligomerize and form minuscule pores in the plasma membrane [36,37,38,39]. However, whether GSDMD pore is the exit site for HMGB1release has yet to be determined. Also, the exact nature of inflammasome and their role in GSDMD cleavage in HMGB1 release in autophagy-deficient hepatocyte has yet to be examined in detail.

## 5. HMGB1 in Liver Pathogenesis

HMGB1, after its release from the stressed or activated cells, exhibits pleiotropic functions, depending upon the cellular or tissue context. In general, extracellular HMGB1 impacts hepatic inflammation, fibrosis, ductular reaction, and hepatic tumorigenesis. In the following sections, we will discuss, in more detail, the role of HMGB1 in liver pathogenesis.

### 5.1. HMGB1 in Liver Inflammation

HMGB1 is the prototypical DAMP that could act as telltales of danger by eliciting early immune responses. HMGB1 alerts antigen-presenting cells of the immune system, which then initiates an adaptive immune response. HMGB1 is also well known to play a key role in sterile (i.e., non-microbial) inflammation, a key process that occurs in common liver diseases such as non-alcoholic steatohepatitis, alcoholic steatohepatitis, and drug-induced liver injury (DILI). Unlike other inflammatory cytokines, such as TNFα and IL-1β, whose cognate plasma membrane receptor families are clearly defined, HMGB1 interacts with several seemingly unrelated receptors such as TLR2, TLR4, and RAGE. TLR4 appears to be the primary receptor for HMGB1 in mediating macrophage activation and cytokine release during tissue injury. Activation of these HMGB1 binding receptors results in a wide range of inflammatory responses, including the production of proinflammatory cytokines and recruitment of immune cells to the site of tissue injury. However, in acute liver injury, such as during acetaminophen overdose, HMGB1 could also escalate the immune cell-mediated liver injury (see below). HMGB1 could also act in a complex with other pro-inflammatory mediators, such as single-stranded DNA, LPS, IL-1β, and nucleosomes, to trigger inflammation. Structurally, among the two-DNA binding domains of HMGB1, the B-box recapitulates the inflammatory activities of the full-length protein, whereas the A-box antagonizes it [40].

### 5.2. HMGB1 in Liver Fibrosis

In response to liver injury, the fibrotic response is activated in the liver as a repair process. HMGB1 has been shown to be a driving factor of hepatic fibrosis. HMGB1 released in response to parenchymal liver damage results in hepatic stellate cells (HSCs) transdifferentiation into scar-forming liver myofibroblasts. Activated HSCs then secrete matrix proteins such as collagen type I in extracellular space, resulting in liver scar formation.

HMGB1 activates HSCs to stimulate liver fibrosis in the in vitro and in rodent models of fibrosis [22,41]. Moreover, in chronic liver diseases, such as long-term hepatitis B virus (HBV) or hepatitis C virus (HCV) infection, primary biliary cirrhosis (PBC), or alcoholic steatohepatitis (ASH), the level of HMGB1 expression is correlated with the fibrosis stage [22,24,42].

In multiple experimental models of mouse liver fibrosis attributed to DILI, cholestasis, ASH, or NASH, elevated expression and release of HMGB1 are induced [24]. In support of this observation, neutralization of HMGB1 protected, whereas injection of recombinant HMGB1 promoted, liver fibrosis [24]. Interestingly, the extracellular HMGB1 appears to activate the RAGE-PI3K-AKT1/2/3 pathway or the RAGE-ERK pathway to upregulate the collagen type I synthesis by the HSC [22,24]. More importantly, HMGB1 does not stimulate the fibrotic signal via TLR2, TLR4, and TLR9. The receptor-specific selectivity for a fibrotic response seems to be distinct from the liver inflammation where HMGB1 interacts with RAGE and TLRs.

HMGB1 also stimulates the HSC migration in vitro and in vivo, a critical event driving the progression of liver fibrosis [24]. How HMGB1 stimulates the HSC migration during fibrosis development is less clear. Thus, HMGB1 release from injured hepatocytes activates a hepatic fibrotic process in a paracrine manner. Interestingly, in the context of autophagy-deficiency in hepatocytes, which presented robust liver injury and fibrosis, *Hmgb1* deletion did not affect liver fibrosis [7]. In a similar manner, in the diethylnitrosamine (DEN)-induced carcinogenesis model, HMGB1 also did not show any remarkable effect on liver fibrosis [43]. Thus, HMGB1 appears to play a key role in liver fibrosis by activating the HSC via RAGE receptor to stimulate collagen synthesis and deposition in liver fibrosis in the particular rodent models of fibrosis.

### 5.3. HMGB1 in Ductular Reaction

The term “ductular reaction” was coined by Popper et al. in 1957 [44]. Ductular reaction (DR) implies a reaction of a ductular phenotype that encompasses the complex of stroma, inflammatory cells, and other structures of diverse systems, all of which participate in the reactive lesion. Various types of liver injury promote DR, characterized by the periportal accumulation of small round cells referred to as ductular cells (DCs) or oval cells or Hepatic progenitor cells (HPCs).

Histologically, DCs appear as isolated small cuboidal cells or strings of such cells with scant cytoplasmic organelles. Currently, these DCs are identified using various markers—Cytokeratin (CK) 19, CK7, OV-6, A6, CD24, CD133, Neuro-Glia Antigen2(NGA2), epithelial cell adhesion molecule (EpCAM), sex-determining region Y-Box 9 (SOX9), trophoblast cell surface antigen 2 (TROP2), and leucine-rich repeat containing G protein-coupled receptor 5 (LGR5). In human chronic liver diseases, there is an association between the severity of liver disease and the increasing number of DCs [45]. For example, the fibrosis stage and ductular cells are strongly correlated. Hepatic DCs are increased significantly in patients with genetic hemochromatosis, alcoholic liver disease, non-alcoholic fatty liver disease, chronic hepatitis C or PBC, primary sclerosing cholangitis (PSC), biliary atresia (BA), or acetaminophen toxicity [45,46,47].

At present, the exact origin of DCs and the associated signaling pathway for the DCs generation is unclear. However, the combination of oxidative liver damage and inhibited replication of mature hepatocytes has been linked to an increasing number of hepatic DCs. We recently reported the important role of HMGB1 in the expansion of DCs in autophagy-deficient livers and in various toxic dietary models, such as 3,5-diethoxycarbonyl-1,4-dihydrocollidine–supplemented (DDC-supplemented) diet-fed mice, and choline-deficient, ethionine-supplemented (CDE) diet-fed mice [7]. DR, as measured by the level of CK19 or A6 or by H-E staining, was significantly suppressed in *Hmgb1* deleted liver in these models. We also showed that pharmacological inhibition of HMGB1 release via ethyl-pyruvate treatment prevented DR. These results confirmed that HMGB1 was required for the expansion of DCs in the autophagy-deficient and toxic dietary model. These DCs well expressed RAGE, and HMGB1 acted extrinsically via RAGE to promote DR. The decreased expansion of DCs has also been reported in *Rage*-ablated *Mdr2-/-* mice [48]. Similar observations on HMGB1′s role in DC expansion had been reported in DEN-induced hepatic carcinogenesis [43]. Mechanistically, HMGB1 activated the ERK signaling pathway via RAGE to increase the proliferation of DCs, suggesting that extracellular HMGB1 plays a mitogenic role during DR.

Similar to HMGB1, some other hepatic factors such as a TNF family member called TWEAK (TNF-like weak inducer of apoptosis), connective tissue growth factor (CTGF/CCN2), pleiotrophin, fibroblast growth factor 10 (FGF10), and fibroblast growth factor 7 (FGF7) have been implicated in DCs’ proliferation [49,50,51,52,53]. TWEAK was found to stimulate ductular cell proliferation in mouse liver through its receptor Fn14 [50]. Transgenic mice overexpressing TWEAK in hepatocytes exhibit periportal DCs hyperplasia. Likewise, FGF7 was found to be another critical regulator of DCs [51]. FGF7 expression was induced concomitantly with the ductular response in the liver of mouse models of liver injury. Moreover, FGF7-deficient mice exhibited marked depressed DCs expansion [51]. It is possible that HMGB1-RAGE signaling could cross-talk with the TWEAK-Fn14 signaling to cause the DCs expansion.

### 5.4. HMGB1 in Liver Tumorigenesis

Hepatic tumorigenesis develops in the context of chronic cell death, inflammation, fibrosis, and maladaptive repair and regenerative response. As one potential mediator of these pathological features, the role of HMGB1 in tumor development and progression is apparent. Supporting this concept, the circulating level of HMGB1 was found to be highly elevated in patients with HCC and significantly correlated with tumor size and other clinicopathological features [54,55]. A higher expression of HMGB1 has also been observed in human liver cancer tissues compared with the normal tissue [55,56] and with different HCC cell lines [55].

The tumorigenic role of HMGB1 is further supported by studies in mouse models of hepatocarcinoma (HCC) that arise in response to carcinogens such as diethylnitrosamine (DEN) and carbon tetrachloride (CCl4) that could induce chronic hepatocyte death, inflammation, and fibrosis. Deletion of *Hmgb1* inhibited tumor development in the DEN-induced HCC model [43]. Similarly, we reported that in an autophagy-deficiency-related hepatic tumorigenesis, HMGB1 plays a vital role in tumor formation [7]. Genetic ablation of *Hmgb1* delayed the tumor development in autophagy-deficient livers. HMGB1 can mediate the tumor development via RAGE receptor downstream of its release because codeletion of RAGE delayed the tumor development in autophagy-deficient livers [7]. Furthermore, the data support the notion that HMGB1 release is necessary and it has to act through RAGE extrinsically to promote liver tumor. Interestingly, HMGB1 did not exert any major impact on hepatic inflammation or fibrosis or hepatic injury in these models [7,43]. In both of these studies, HMGB1 deletion reduced the expansion of progenitor cells (DCs), a key feature of hepatic tumorigenesis [7,43]. It is unknown whether HMGB1 mediated DCs expansion could directly contribute as a cellular source for the tumor formation. Alternatively, these DCs could modulate the tumor microenvironment by secreting vasculogenic factors such as vascular endothelial growth factor D (VEGF-D), platelet-derived growth factor C (PDGFC), and angiopoietin 1 (ANGPT1) [57].

The role of HMGB1 in tumorigenesis could also be mediated by a cell-intrinsic mechanism [58]. In DEN-induced HCC model, HMGB1 was found to transcriptionally regulate the expression of yes-associated protein (YAP), a major downstream effector of the hippo pathway that contributes to liver tumorigenesis by inducing hypoxia-inducible factor 1α (HIF1α)-dependent aerobic glycolysis [58]. In a similar line, intracellular HMGB1 has also been shown to facilitate Peroxisome proliferator-activated receptor γ (PPARγ) coactivator 1α (PGC1α) expression and mitochondrial biogenesis in hypoxic HCC cells, hence promoting tumor survival and proliferation [59].

Thus, HMGB1 is a protumorigenic factor in the context of liver diseases. Extracellularly, HMGB1 promotes and sustains inflammatory or fibrotic microenvironment that results in hepatic tumor cell proliferation. Intracellularly, HMGB1 could modulate the metabolic function of the hepatic tumor cells, through regulating YAP/glycolysis and PGC-1α/mitochondrial biogenesis. Regardless of the underlying mechanism by which HMGB1 promotes hepatocarcinogenesis, these studies suggest that HMGB1 could be a potential therapeutic target for anticancer therapy. Glycyrrhizin, a phytochemical that directly inhibits HMGB1 [60], is widely used for the treatment of liver disease in Asia, and there are indications that glycyrrhizin can reduce HCC development in certain patient subsets [61].

### 5.5. HMGB1 in Liver Regeneration

Tissue damage causes inflammation and tissue regeneration. How tissue damage-related repair and regenerative is linked to inflammation is less clear [62]. HMGB1 appears to coordinate tissue repair and inflammation by switching among alternative redox forms. HMGB1 containing a disulfide bond (oxidized) is proinflammatory molecule, whereas fully reduced HMGB1 can accelerate liver repair and regeneration [63]. Pharmacological treatment with fully reduced HMGB1 (oxidative resistant) variant can accelerate regeneration without exacerbating inflammation in an acute liver injury condition [63]. Systemic administration of fully reduced HMGB1 increased the hepatocyte proliferation significantly earlier than the control mice. Interestingly, administration of AMD3100, an inhibitor of CXCR4 receptor, blocked this regenerative effect of HMGB1 [63]. Thus, the regenerative effect of fully reduced HMGB1 promotes liver regeneration through the CXCR4 receptor. Interestingly, the fully reduced HMGB1 binds inefficiently to TLR4 or RAGE. A similar observation has been made in acute muscle injury models [63]. In contrast, HMGB1 did not show any effect in the two-third partial hepatectomy or CCL4 liver injury model of liver regeneration [43]. Hepatocyte-specific HMGB1-deleted mice, when subjected to two-third partial hepatectomy or single injection of CCl4, did not show any effect in the regulation of hepatocyte proliferation in either model [43]. This indicates that the HMGB1 can function as a mediator of liver regeneration, but in a disease context-dependent manner.

## 6. HMGB1 Has a Pathogenic Role in Common Liver Diseases

HMGB1 plays a key role in liver inflammation, fibrosis, ductular reaction, and hepatic tumorigenesis, as discussed above. We will next describe below the role of HMGB1 in the context of common liver diseases such as non-alcoholic fatty liver disease (NAFLD), alcoholic liver disease (ALD), and drug-induced liver injury (DILI).

### 6.1. HMGB1 Participates in the Pathogenesis of Non-Alcoholic Fatty Liver Disease

HMGB1 plays a critical role in initiating and maintaining a chronic inflammatory state in the liver tissue. Since liver inflammation is one of the histological markers that pathologically delineate simple steatosis from non-alcoholic steatohepatitis (NASH), HMGB1 could be a hepatic factor mediating the progression of steatosis to NASH. In NAFLD, lipotoxicity causes the release of HMGB1 and drives sterile inflammation [64]. In support of this notion, a large cohort study of pediatric patients with biopsy-proven NAFLD showed higher levels of circulating HMGB1 in children with NAFLD than obese-only controls. The circulating HMGB1 level was not only correlated with the degree of fibrosis, but also with the levels of inflammatory mediators, such as TGF-β and MCP-1 [65]. These findings suggest the critical role of HMGB1 in the progression of NAFLD [65]. Hence, serum level of HMGB1 has been considered to be a potential biomarker for early diagnosis of NAFLD and a therapeutic target for prevention and intervention of NAFLD-associated inflammation.

Increased expression and release of hepatic HMGB1 also occurred in the early phase of murine nutrient-excess models of NAFLD such as that caused by high-fat diet (HFD) or high-fat diet with high cholesterol and a high sugar supplement diet (HF-HC-HSD) [66,67]. The upregulated expression of HMGB1 was mediated by microRNA (miR)-429 and JNK1/JNK2-ATF2 signaling axis [66]. The extracellular HMGB1 then triggered TLR4/MyD88 signaling in hepatocytes and inflammation in response to free fatty acid infusion or HFD. Moreover, TLR4 deletion in hepatocytes prevented obesity-induced insulin resistance and systemic inflammation during HFD feeding. Interestingly, the circulating level of HMGB1 due to release from steatotic livers can also trigger inflammation in the intestine. HMGB1 can bind to RAGE receptors in the distal intestine and activates TLR4 translocation to lipid raft via NADPH oxidase complex assembly [68].

HMGB1 released from the steatotic liver also works through the RAGE receptor. Genetic deletion of RAGE prevented the effect of HFD on energy expenditure, weight gain, adipose tissue inflammation, and insulin resistance. Treatment with soluble RAGE partially protected against HFD-induced inflammation and weight gain [69]. Functional inhibition of HMGB1 by a neutralizing antibody in vivo also mitigated HFD-induced liver damage, steatosis, inflammation, and liver function impairment [70]. On the other hand, genetic deletion of HMGB1 in hepatocytes rapidly promoted HFD-induced weight gain and obesity, with enhanced hepatic fat deposition (steatosis). Mechanistically, HMGB1 was correlated to the enhanced ER stress-associated impaired mitochondrial oxidative phosphorylation and impaired FFA β-oxidation [71]. These findings suggest that HMGB1 in NAFLD not only acts as an alarmin signal, but also plays an important role in lipid metabolism. HMGB1 is also necessary for maintaining proper mitochondrial function, β-oxidation, and preventing ER stress. It is possible that cytosolic HMGB1, during the process of release, partake in the mitochondrial metabolic function, whereas the extracellular HMGB1 acts as an alarmin for communicating the lipid stress signaling to the neighboring hepatic cells.

Finally, at the cellular level, similar to the in vivo data, hepatic cells treated with FFA, such as saturated palmitic acid, also rapidly induce the expression and extracellular release of HMGB1 [66]. Blocking of HMGB1 by small interfering RNA (siRNA) or pharmacological such as by salvianolic acid B (SalB) also protected hepatocytes against FFA-induced TNFα and IL-6 production [70]. Such an effect was found to be mediated by sirtuin 1 (SIRT1)-mediated deacetylation of HMGB1 proteins that cause HMGB1 retention in the nucleus during HFD [70].

Nonetheless, how HMGB1 expression and release (active or passive) is enhanced in NAFLD is less clear, and whether the extracellular HMGB1 affects the downstream liver pathologies other than inflammation, such as hepatic fibrosis or ductular reactions, has not been examined (Figure 3).

### 6.2. HMGB1 Participates in the Pathogenesis of Alcoholic Liver Disease

The spectrum of alcoholic liver disease (ALD) ranges from fatty liver to steatohepatitis, progressive fibrosis, cirrhosis, and hepatocellular carcinoma. Hepatic inflammation is a key component contributing to the pathogenesis of ALD. Since HMGB1 is an early proinflammatory cytokine upregulated in response to sterile and non-sterile tissue injury, it may play critical roles in the pathogenesis of ALD. Liver biopsies from patients with acute alcoholic steatohepatitis (ASH) superimposed on ALD and cirrhosis showed a significant and progressive increase in HMGB1 expression and translocation from the nucleus to the cytoplasm [72]. Moreover, HMGB1 induction and translocation were correlated with the disease stage in human ALD [72]. Similarly, in rodent models of ALD, alcohol intake elevated expression of HMGB1, nucleocytoplasmic shutting, and secretion from the hepatocyte. The ROS generated during ethanol metabolism appears to regulate HMGB1 release. Treatment of antioxidants in ALD prevented HMGB1 release [72]. It is plausible that increased ROS generated by alcohol metabolism cause post-translational modification such as acetylation of HMGB1 to stimulate its active release. The extracellular HMGB1 during the ALD progression could then orchestrate the immune cell recruitment and their activation to secrete inflammatory cytokines. Interestingly, genetic ablation of *Hmgb1* protected the mice from both alcoholic steatosis and liver injury, suggesting that HMGB1 has pathogenic role in ALD.

### 6.3. HMGB1 in Drug-Induced Liver Injury

Acetaminophen (APAP) hepatotoxicity is the leading cause of drug-induced liver injury (DILI) in the United States and other industrialized nations [73]. When ingested in excess, APAP is converted to its reactive toxic metabolite called *N*-acetyl-p-benzoquinone imine (NAPQI) by hepatic cytochrome Cyp2e1 (to a lesser extent by Cyp1A2 and 3A4). NAPQ1 then covalently binds to critical cellular proteins, especially the mitochondrial proteins, to cause centrilobular necrosis [74]. The necrotic hepatocytes then release DAMPs such as HMGB1 [75].

APAP overdose significantly increases the circulating level of HMGB1 [76,77]. Reflecting this clinical observation, APAP intoxication in mice also increased the level of circulating HMGB1 [75,78]. In APAP overdose, massive necrotic hepatocytes passively release HMGB1. The extracellularly released HMGB1 not only induces the infiltration of immune cells such as neutrophils, but also stimulates immune cells-mediated liver injury [75]. The extracellularly released HMGB1 activates TLR4-IL23 pathway in macrophages to generate the IL-15-producing γδT cells, which then mediates neutrophil infiltration [79]. The extracellularly released HMGB1 has been reported to bind to complement C1q and activates the classical complement pathway in APAP-mediated hepatotoxicity [80].

HMGB1 uses RAGE as its receptors to recruit neutrophils, but not macrophages, into necrotic tissues during APAP toxicity [75]. The HMGB1/RAGE-induced neutrophil recruitment further escalates liver injury. How infiltrated neutrophils escalate liver injury is not clear. It has been suggested that the TIR domain-containing adapter-inducing interferon-β(TRIF)/receptor-interacting protein (RIP) 3 kinase axis-mediated necroptosis plays a critical role for the deleterious feed-forward liver injury process [81]. Hence, HMGB1 not only plays a role in neutrophil migration, but also assists neutrophil amplification of liver injury following necrosis (immune-mediated liver injury). Remarkably, hepatocyte-specific HMGB1 ablation or pharmacologically treating HMGB1 by glycyrrhizin rescued APAP-induced hepatic inflammation and injury [75,82]. Blockade of HMGB1 using a neutralizing antibody has also been shown to improve hepatocyte regeneration and liver structure recovery in APAP toxicity [83]. Thus, acetaminophen overdose causes acute liver inflammation with neutrophil infiltration. The extracellularly released HGMB1 by necrotic hepatocytes triggers inflammation and amplifies APAP-induced liver injury, possibly through feed-forward TRIF/RIPK3-dependent hepatocyte necrosis.

## 7. Therapeutic Potential of HMGB1

Extracellular HMGB1 is elevated in various preclinical and clinical models, to drive the pathogenesis of acute or chronic liver diseases. Given its widespread expression as well as that of its receptors in liver, targeting the HMGB1 ligand or its receptor represents an important potential application in liver disease therapeutics. Studies in preclinical animal models using HMGB1 antagonists also clearly supports the notion that blocking excessive amounts of extracellular HMGB1 offers an attractive therapeutic target to ameliorate inflammatory liver diseases. Various small molecules (glycyrrhizin, ethyl pyruvate, salicylic acid, heparin, inflachromene, epigallocatechin 3-gallate) or peptide-based (A-box, R2F8, S100-based peptides, soluble RAGE) antagonists have been proven to directly bind HMGB1 or HMGB1–receptor complex and attenuate the HMGB1 downstream signaling events [84]. On the contrary, the therapeutic intervention to regulate intracellular HMGB1 biology must still wait for a deeper understanding of endogenous modulation of HMGB1 activity and signaling. Nevertheless, further basic and clinical studies on various HMGB1 isoforms and HMGB1 protein complexes in the context of liver diseases are warranted to identity means to exploit this therapeutically.

## 8. Conclusions

HMGB1 has pleiotropic functions, depending upon its cellular location. While intracellular HMGB1 is required for DNA architectural maintenance, gene expression, and induction of autophagy or mitophagy, extracellular HMGB1 critically participates in liver inflammation, fibrosis, ductular reaction, and hepatic tumorigenesis. Because of this functional diversity, HMGB1 has been implicated in the pathogenesis of many common liver diseases and is being considered as a potential therapeutic target.

## Figures and Tables

**Figure 1 ijms-20-05314-f001:**
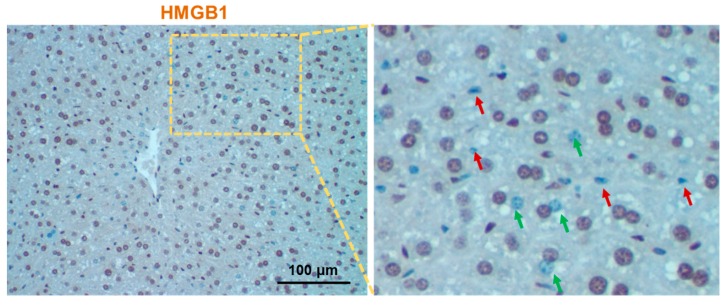
High-mobility group box1 (HMGB1) immunohistochemistry of normal mouse liver section. Liver section of normal mouse was immunostained with anti-HMGB1 and images were digitally acquired using Nikon Eclipse E200 microscope. The right panel is the enlarged image of the left panel, where red and green arrows indicate the non-parenchymal and parenchymal cells devoid of nuclear HMGB1, respectively.

**Figure 2 ijms-20-05314-f002:**
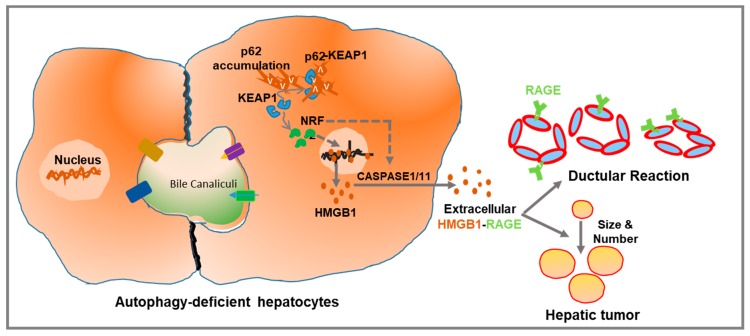
Schematic representation of the mechanism of HMGB1 release and its pathological impact in the autophagy-deficient liver. Autophagy deficiency causes accumulation of p62, which can physically associate with KEAP1 to activate the antioxidative transcription factor, NRF2. Activation of NRF2 causes HMGB1 release via CASPASE 1/11-mediated inflammasomes. Extracellular HMGB1, via receptor for advanced glycation end products (RAGE), promotes ductular reaction and tumor development without affecting liver inflammation and fibrosis. Dotted lines indicate possible but unproven processes.

**Figure 3 ijms-20-05314-f003:**
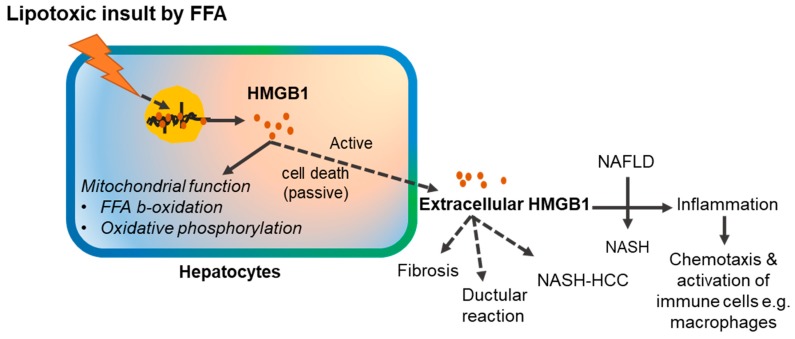
Role of HMGB1 in the pathogenesis of non-alcoholic fatty liver disease (NAFLD). Solid lines indicate processes with experimental supports. Dotted lines indicate possible but unproven processes. FFA: Free fatty acid; NASH; Non-alcoholic steatohepatitis; NASH-HCC: Non-alcoholic steatohepatitis-associated hepatocellular carcinoma.

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
