# Peer review of "Role of High-Mobility Group Box-1 in Liver Pathogenesis"

_ijms, 2019, doi:10.3390/ijms20215314_

Round 1

Reviewer 1 Report

Title: Role of High-Mobility Group Box-1 in Liver Pathogenesis

In this manuscript, the authors reviewed the multifaceted pathophysiological roles of high-mobility group box 1 (HMGB1) on the liver diseases. Discussion on the HMGB1 functions related to its cellular localizations are comprehensive and clear. Overall, this review is well written and provides useful summary and perspectives on the HMGB1. There are only a few critiques listed below to be considered.

Critiques for revision:

For figure 1, scale bars should be provided. References are needed for sentences between lines 101-102. The role of HMGB1 on the cell surface is suggested to be expanded. It is curious about the HMGB1’s function on liver regeneration or transplantation. Also, what is the therapeutic potential or implication for HMGB1?

Author Response

Reviewer 1

Title: Role of High-Mobility Group Box-1 in Liver Pathogenesis

In this manuscript, the authors reviewed the multifaceted pathophysiological roles of high-mobility group box 1 (HMGB1) on the liver diseases. Discussion on the HMGB1 functions related to its cellular localizations are comprehensive and clear. Overall, this review is well written and provides useful summary and perspectives on the HMGB1. There are only a few critiques listed below to be considered.

Response: We thank the reviewer for the positive critiques.

Critiques for revision:

For figure 1, scale bars should be provided.

Response: We have provided the scale bar for Figure 1.

References are needed for sentences between lines 101-102.

Response: We have provided related references.

The role of HMGB1 on the cell surface is suggested to be expanded.

Response: According to the reviewer’s suggestion, we have briefly described the role of HMGB1 in the cell surface as most of the cell surface role of HMGB1 has been reported in the non-hepatic cells such as cord blood mononuclear cells or neurons. We included this in section 2, page 2, and Line 55-57.

It is curious about the HMGB1’s function on liver regeneration or transplantation.

Response: According to the reviewer’s suggestion, we have described the role of HMGB1 in liver regeneration in section 5.5, page 8, and line 345-362.

Also, what is the therapeutic potential or implication for HMGB1?

Response: According to the reviewer’s suggestion, we have described the therapeutic potential of HMGB1 in section 7, page 11, and line 468-482.

Reviewer 2 Report

Khambu and colleagues reviewed the role of HMGB1 in liver pathogenesis. On the whole, the review is adequately written, with a fine scope, focus and flow. The following points are for the authors considerations.

1. Is Fig 1 necessary, and if so what value exactly does it add to this review? In any case, Fig 1's origin is not properly cited and the description in the legend is inadequate. How was HMGB1 visualized, and what exactly is the significance of HMGB1-low cells? Are these sections from wild-type mouse?
2. Granted that unconventional secretion of HMGA1 is not well understood, the authors' description should not add to the confusion. Line 118-19 states that it is secreted 'as a soluble protein', but then 'secretary autophagy' is mentioned in line 151, and Fig 3 showed EVs. The confused reader would wonder if secretory autophagy could occur without Atg7 and how would HBGB1 contained within EV membranes bind to its cognate receptors.
3. Acetylation status of HBGB1 is mentioned but the role for deacetylation and stress-induced deacetylase Sirt1 in its release/secretion is known but not given enough coverage.
4. Fig 2 is rather plain on the hepatic tumor part. A better illutrated scheme by which HMGB1 leads to tomorigenesis would be helpful.
5. Line 197-198 - The TLRs could hardly be described as 'seemingly unrelated'.
6. HMGB1 as a biomarker seems clear, but what about it being a therapeutic target? This is mentioned several times in the text but there is no description of how HBGB1 could be targeted (compounds, Abs, RNA-based silencing?).
7. The text would benefit from another round of language editing. Sentence in line 102-03, for exmple, is awkward. There are also some typographical errors.

Author Response

Reviewer 2

Khambu and colleagues reviewed the role of HMGB1 in liver pathogenesis. On the whole, the review is adequately written, with a fine scope, focus and flow.

Response: We thank the reviewer for the positive critiques.

The following points are for the authors considerations.

1. Is Fig 1 necessary, and if so what value exactly does it add to this review? In any case, Fig 1's origin is not properly cited and the description in the legend is inadequate. How was HMGB1 visualized, and what exactly is the significance of HMGB1-low cells? Are these sections from wild-type mouse?

Response: Figure 1 is presented to emphasize the two messages 1)Not all hepatic cells express the nuclear HMGB1, the reason is unknown to date. 2) Those normal-looking, HMGB1 devoid hepatic cells may have compensatory mechanisms for the lack of HMGB1.

Additional citations have been included and the description in figure legend has been elaborated for clarity. The image scale bar has also been included.

Granted that unconventional secretion of HMGB1 is not well understood, the authors' description should not add to the confusion. Line 118-19 states that it is secreted 'as a soluble protein', but then 'secretary autophagy' is mentioned in line 151, and Fig 3 showed EVs. The confused reader would wonder if secretory autophagy could occur without Atg7 and how would HBGB1 contained within EV membranes bind to its cognate receptors.

Response: We have modified Figure 3 to increase clarity.

Acetylation status of HBGB1 is mentioned but the role for deacetylation and stress-induced deacetylase Sirt1 in its release/secretion is known but not given enough coverage.

Response:  According to the reviewer’s suggestion, we have described the role of deacetylation mediated by Sirt 1 and HDAC1/4 in HMGB1 release in section 3.0, page 4, and line 152-160.

Fig 2 is rather plain on the hepatic tumor part. A better illustrated scheme by which HMGB1 leads to tumorigenesis would be helpful.

Response: We have modified Figure 2 to demonstration that extracellular HMGB1 increased tumor size and number in the autophagy-deficient liver.

Line 197-198 - The TLRs could hardly be described as 'seemingly unrelated'.

Response: We have deleted the TLR9 from the list of HMGB1 receptors, which has not been reported to play a role in liver inflammation. TLRs are evolutionally conserved type 1 transmembrane proteins localized either at the plasma membrane (TLRs 1, 2,4-6, and 10) or within the endosomal compartment (TLRs 3 and 7-9), protecting the host against threats present in either the extracellular or intracellular environment (Akira and Takeba, 2004, PMID:15229469). Among the various TLRs, TLR4 is the best-characterized HMGB1 receptor for macrophage activation and initiation of pro-inflammatory signaling.

HMGB1 as a biomarker seems clear, but what about it being a therapeutic target? This is mentioned several times in the text but there is no description of how HBGB1 could be targeted (compounds, Abs, RNA-based silencing?).

Response: According to the reviewer’s suggestion, we have described the therapeutic potential of HMGB1 in section 7, page 11, and line 468-482.

The text would benefit from another round of language editing. Sentence in line 102-03, for example, is awkward. There are also some typographical errors.

Response: Sentence in line 102-3(now 115-116) has been modified to increase clarity. Other typographical errors have also been corrected and highlighted as red in the text file.

Round 2

Reviewer 1 Report

The authors have revised the manuscript and addressed all the comments appropriately.